# Pairwise CNN-Transformer Features for Human–Object Interaction Detection

**DOI:** 10.3390/e26030205

**Published:** 2024-02-27

**Authors:** Hutuo Quan, Huicheng Lai, Guxue Gao, Jun Ma, Junkai Li, Dongji Chen

**Affiliations:** 1College of Computer Science and Technology, Xinjiang University, Urumqi 830017, China; quanhutuo@stu.xju.edu.cn (H.Q.); gaoyangshuang123@stu.xju.edu.cn (G.G.); majun@stu.xju.edu.cn (J.M.); lijunkai@stu.xju.edu.cn (J.L.); 107552101245@stu.xju.edu.cn (D.C.); 2Xinjiang Key Laboratory of Signal Detection and Processing, Xinjiang University, Urumqi 830017, China

**Keywords:** computer vision, human–object interaction detection, convolutional neural network, transformer, feature fusion

## Abstract

Human–object interaction (HOI) detection aims to localize and recognize the relationship between humans and objects, which helps computers understand high-level semantics. In HOI detection, two-stage and one-stage methods have distinct advantages and disadvantages. The two-stage methods can obtain high-quality human–object pair features based on object detection but lack contextual information. The one-stage transformer-based methods can model good global features but cannot benefit from object detection. The ideal model should have the advantages of both methods. Therefore, we propose the Pairwise Convolutional neural network (CNN)-Transformer (PCT), a simple and effective two-stage method. The model both fully utilizes the object detector and has rich contextual information. Specifically, we obtain pairwise CNN features from the CNN backbone. These features are fused with pairwise transformer features to enhance the pairwise representations. The enhanced representations are superior to using CNN and transformer features individually. In addition, the global features of the transformer provide valuable contextual cues. We fairly compare the performance of pairwise CNN and pairwise transformer features in HOI detection. The experimental results show that the previously neglected CNN features still have a significant edge. Compared to state-of-the-art methods, our model achieves competitive results on the HICO-DET and V-COCO datasets.

## 1. Introduction

Human–object interaction (HOI) detection aims to recognize interactions between humans and objects in a scene. This task requires the precise localization of human–object pairs and the identification of associated actions. HOI detection can facilitate the development of many tasks, such as anomaly detection [1], image understanding [2], and action recognition [3]. HOI detection methods [4] can be categorized into two-stage methods [5,6,7], traditional one-stage methods [8,9,10], and end-to-end one-stage methods [11,12]. Two-stage methods leverage an object detector to obtain confidence scores for humans and objects, subsequently filtering out low scores while retaining high-quality instances. However, these methods face challenges in capturing robust contextual cues. Traditional one-stage methods improve inference speed by parallelizing object detection and interaction prediction. However, matching human–object pairs and interactions requires complex strategies. Inspired by DEtection TRansformer (DETR) [13], the end-to-end one-stage methods maintain a concise architecture and avoid the matching process of human–object pairs and interactions. These methods leverage global features from the transformer [14], providing the model with rich contextual information. However, they cannot acquire prior knowledge from object detection. Based on the analysis above, our goal is to design a unified model that combines the advantages of both methods. The model can take advantage of the object detector and also has rich contextual information.

Determining how to use the object detector effectively. We note that feature pyramid network (FPN) [15] and HRNet [16] successfully enhance the representation of features by fusing low-resolution features with high-resolution features. Inspired by this, we explore using an object detector to extract multiple types of features. This aims to enhance the features representation.

Current one-stage methods [11,17] outperform two-stage methods [18,19] across the board, so many studies ignore the potential of two-stage methods. Recently, UPT [20] analyzes why one-stage methods are superior to two-stage methods. It is mainly due to using more efficient object detectors and transformer representation capabilities. The two-stage method SCG [21] achieved a significant performance improvement by replacing Faster R-CNN [22] with DETR [13] (object detector). Its performance is close to that of the one-stage method QPIC [11]. Inspired by UPT, our model selects DETR as the object detector. Since DETR inspired a series of one-stage transformer-based methods, these methods use both the convolutional neural network (CNN) backbone and the transformer architecture [14]. However, existing research focuses on using the transformer features to predict HOIs [11,17,23,24,25] while ignoring the CNN features. Thus, the intuitive scheme is to fuse CNN and transformer features.

Determining how to obtain contextual cues. The interaction head of the two-stage methods [20,26] uses a transformer encoder structure, where the input is only the human–object pair features. The one-stage methods [11,17] use a transformer decoder structure, where the inputs are global features and learnable queries. In two-stage methods, the simple concatenation of human and object features lacks sufficient contextual information. However, the global features from the transformer have proven effective in one-stage methods. Therefore, we propose introducing these global features to provide more explicit contextual cues. In addition, introducing global features can be used to visualize the attention of the model, facilitating future research.

To achieve the above goals, we propose the Pairwise CNN-Transformer (PCT), a simple and intuitive two-stage detector that utilizes pairwise CNN and pairwise transformer features. Figure 1 illustrates our PCT pipeline:We extract features of human and object instances from both the CNN backbone and the transformer architecture.The instance features are fused and paired to form enhanced human–object pair features.The enhanced features are fed into the interaction head with global features to predict actions.

Determining why previous work did not compare transformer features and CNN features. Since one-stage transformer-based methods [11,17] use a set of queries to predict HOIs, the number of learnable action queries is fixed. Two-stage methods [5,6,20] generate human–object pairs by pairwise matching. The number of human–object pairs is variable. Different selection strategies for human–object pairs make it challenging to compare the performance of CNN and transformer features fairly. Although UPT [20] has the conditions to compare them, it does not investigate this issue. To fairly compare their performance in HOI detection, we construct three variant models of UPT. Experimental results show that CNN features outperform transformer features in three sets of comparative experiments.

In summary, our contributions are as follows:We propose fusing the CNN and transformer features of the instances to enhance the feature representations. These enhanced representations effectively improve the precision of HOI detection while introducing little computational cost.We propose utilizing global features from the transformer to enrich contextual information. Through cross-attention, human–object pair features can aggregate valuable information from these global features.We fairly compare the performance of pairwise CNN and pairwise transformer features in HOI detection. The results show that CNN features outperform transformer features.Our model achieves competitive performance on two commonly used HOI datasets, HICO-DET and V-COCO.

We propose a simple and efficient two-stage method PCT. Compared with the recent two-stage method UPT, our method performs better in terms of accuracy, trainable parameters, and inference time. In addition, two-stage methods have a shorter training time and smaller training memory compared to one-stage methods. The code is available at https://github.com/hutuo1213/PCT (accessed on 25 January 2024).

## 2. Related Work

### 2.1. Object Detection

The object detection task requires locating and identifying objects. This task is the cornerstone of understanding high-level semantics. In recent years, object detection has rapidly evolved from two-stage methods, such as Faster R-CNN [22], to end-to-end methods, such as DETR [13]. Traditional HOI detection methods acquire accurate human and object instances through object detection. Both object detection and HOI detection involve localization and recognition tasks, which makes it possible to migrate object detection strategies to HOI detection. In summary, object detection continues to drive the development of HOI detection. Our method employs a more advanced DETR as the object detector, which consists of a CNN backbone and a transformer network. This method simplifies the detection process by transforming object detection into a set prediction problem.

### 2.2. Features Fusion

Effective feature fusion enhances the expressive power of features. FPN [15] and HRNet [16] fuse low-resolution features with high-resolution features to enhance features. MSFIN [27] obtains rich information by fusing features from different scales. Kansizoglou et al. [28] explore training strategies for multi-modal fusion. We have observed that explicit methods for features fusion have not been proposed in HOI detection. Therefore, we explore how to fuse features to enhance their representations.

### 2.3. Human–Object Interaction Detection

HOI detection can be divided into two-stage and one-stage methods. Two-stage methods first perform object detection and then interaction prediction, while one-stage methods simultaneously perform object detection and interaction prediction. HO-RCNN [5], typically a two-stage CNN-based method, proposes using a multi-stream architecture to apply spatial information. The network uses an object detector to obtain spatial information from human–object pairs to construct spatial features. Subsequent studies have constructed different feature priors based on object detection, including visual features of human–object pairs [7], word embeddings of objects [24,29,30], human pose features [19], and body part features [31,32]. ICAN [6] uses an attention module to aggregate HOI cues selectively. MLCNet [19] combines almost all available feature priors to predict HOI. In addition, graph neural networks have been used to reason about relationships between humans and objects [18,30]. One-stage CNN-based HOI methods [8,9,10] directly predict interactions, increasing the speed of model inference. However, they must match human–object pairs and interactions, resulting in a more complex design.

Inspired by DETR [13], one-stage transformer-based methods are beginning to grow in popularity. Traditional two-stage methods use pairwise matching of humans and objects to predict HOIs. One-stage transformer-based methods directly predict HOIs using a set of queries. QPIC [11] modifies DETR for HOI detection tasks. HOTR [12] uses interaction representations to infer relevant human–object pairs. DOQ [24] performs knowledge distillation based on QPIC. CDN [17] avoids the challenges of multi-task learning through cascading object detection and action classification modules. GEN-VLKT [25] employs the powerful language–image model CLIP [33] to guide feature learning. These one-stage methods utilize the transformer’s global features to obtain good performance in HOI detection. So, the global features contain rich interaction cues.

In our work, we combine the advantages of the two-stage and one-stage methods. Like UPT [20] and STIP [34], our method is a two-stage transformer-based method. However, our research focuses on improving the expressiveness of features, while UPT and STIP are more focused on model design.

## 3. Method

The HOI detection task is to accurately locate humans and objects in a scene and identify action categories among them. The principle of our method is based on the Information Gain (IG) in information theory. Specifically:(1)Gain(P,F)=H(P)−H(P|F)
where *P* is the prediction set of the model, and *F* is the interaction features. Gain(P,F) denotes the IG corresponding to the performance in HOI detection. H(P) and H(P|F) denote information and conditional entropy, respectively. H(·) corresponds to the network model. We can reduce the prediction uncertainty (H(P|F) decreases) by enhancing the expression of the interaction feature *F*. Ultimately, a larger IG is obtained. Section 3.1 introduces our PCT framework. Section 3.2 and Section 3.3 describe the PCT’s two critical components. Section 3.4 presents the training and inference details of the PCT.

### 3.1. Overview

Figure 2 shows our PCT framework. The weights of DETR are frozen throughout the entire training process. In the DETR [13] part, the image $i\in R^{3\times H\times W}$ is input to the CNN backbone to generate feature maps $z\in R^{2048\times \frac{H}{32}\times \frac{W}{32}}$. After adding positional encoding, the feature map *z* performs self-attention (transformer encoder) to model global features $G\in R^{(\frac{H}{32}\times \frac{W}{32})\times 256}$. Subsequently, the global features *G* and learnable object queries $q\in R^{100\times 256}$ perform cross-attention (transformer decoder) to generate human and object instances $f\in R^{100\times 256}$. These instances are decoded into object categories and bounding boxes through a multilayer perceptron (MLP). After filtering out instances with low confidence, the retained bounding boxes are used to extract human and object instances from the CNN backbone network. Then, human and object instances are pairwise matched to generate pairwise CNN features and pairwise transformer features. Add these CNN and transformer features to obtain human–object pair features. In the interaction head part, the human–object pairs and global features perform cross-attention (transformer decoder layer) to aggregate contextual information. Next, these human–object pair features perform self-attention (transformer encoder layer) to refine them [20]. The refined features are decoded into action categories by an MLP.

### 3.2. CNN Features for Human-Object Pairs

Despite the success of the transformer in DETR, it has certain limitations in capturing sequential sequences. CNN are more suitable for capturing the local information than transformers. Therefore, we fuse CNN and transformer features to improve the expressiveness of features.

First, we discussed how DETR works [35]. Figure 3a shows the original image. Figure 3b shows the encoder self-attention at a reference point. We observe that the encoder can separate different object instances. Figure 3c is the decoder attention of an object query. The decoder focuses on features at the edges of objects, such as legs and heads, to better distinguish different objects.

Then, we consider how to achieve this goal. Object queries can activate the corresponding object regions in the image. Therefore, the final representations contain object features (transformer features). In addition, the CNN features of objects can be obtained from the region of interest (ROI) in the CNN backbone. Therefore, we can easily align and fuse the CNN and transformer features of instances.

Specifically, according to the detection results of DETR, we first use non-maximum suppression and set a threshold to retain high-score instances. These instances build a set ${\left\{d_{i}\right\}}_{i=1}^{n}$, where $d_{i}=\left(b_{i},s_{i},c_{i},f_{i}\right)$. $b_{i}\in R^{4}$ is the object bounding box, $s_{i}\in \left[0,1\right]$ is the confidence score, $c_{i}\in K$ is the object class, K is the set of object categories, and $f_{i}\in R^{1\times 256}$ is a human or object instance. The transformer features set ${\left\{f_{i}\right\}}_{i=1}^{n}$ comprises n1 humans ($f_{T-h}\in R^{n1\times 256}$) and n2 objects ($f_{T-o}\in R^{n2\times 256}$). Next, we map the bounding boxes b to the last layer of the ResNet [36] backbone to obtain ROI. Subsequently, global average pooling is applied to these regions to generate CNN features. Finally, the dimensions of the CNN and transformer features are aligned. Specifically:(2)$$f_{Ins}\;=\;GAP\left(ROI\left(z,b\right)\right)$$
(3)$$f_{C-h}\;=\;Relu\left(LN_{1}\left(W_{1}\left(f_{Ins-h}\right)\right)\right)\;f_{C-o}\;=\;Relu\left(LN_{2}\left(W_{2}\left(f_{Ins-o}\right)\right)\right)$$
where $z\in R^{2048\times \frac{H}{32}\times \frac{W}{32}}$ denotes the last layer feature map of the ResNet backbone. $b\in R^{n\times 4}$ denotes the object bounding box, ROI denotes the region of interest operation, and GAP denotes the global average pooling operation. $f_{Ins}\in R^{n\times 2048}$ represents instance features of both humans and objects. $W_{1},W_{2}\in R^{2048\times 256}$ denotes the linear layer, $LN_{1},LN_{2}$ denotes the layer normalization, and Relu denotes the Relu function. $f_{C-h}\in R^{n1\times 256}$ and $f_{C-o}\in R^{n2\times 256}$ denote the CNN features of the human and object, respectively. Next, as shown in Figure 4, We pairwise match human and object instances by concatenation. n1 humans and n2 objects will generate m=n1×(n1+n2−1) human–object pairs. Then, the pairwise CNN features and pairwise transformer features are added element-wise. Specifically:(4)$$Z\;=\;\left(f_{T-h}⊕f_{T-o}\right)+\left(f_{C-h}⊕f_{C-o}\right)$$
where ⊕ denotes concatenation, + denotes element-wise addition. $Z\in R^{m\times 512}$ denotes the human–object pair features, where *m* denotes the number of pairs.

### 3.3. Interaction Head

We argue that simply connecting humans and objects does not provide contextual clues. Given that previous studies often use global features to provide interaction cues, we introduce global features in the interaction head. In Section 4.6, we visualize the decoder attention of the interaction head. Human–object pairs are effectively localized and highlighted when global features are applied. Specifically, the global features are extracted from the transformer encoder of DETR [13]. We use a standard transformer decoder layer that performs cross-attention on human–object pair and global features, as follows:(5)$$G\;=\;Relu(LN_{3}\left(W_{3}\left(G\right)\right))$$
(6)output=Dec(Q=Z,K=V=G)
where $W_{3}\in R^{256\times 512}$ denotes the linear layer. The global features $G\in R^{(\frac{H}{32}\times \frac{W}{32})\times 256}$ are updated to $G\in R^{(\frac{H}{32}\times \frac{W}{32})\times 512}$. Dec denotes the transformer decoder layer, and $output\in R^{m\times 512}$ denotes output features.

Inspired by UPT [20], we used a transformer encoder layer to refine the features of human–object pairs. Specifically:(7)output=Enc(output)
where Enc denotes the transformer encoder layer.

Finally, we decode the output into action classification scores. Specifically:(8)$${\tilde{s}}_{k}\;=\;W\left(output\right)$$
${\tilde{s}}_{k}$ denotes the unnormalized action score. *k* is the *k*-th possible action of the human–object pair. *W* denotes the linear layer. For HICO-DET [5], we use $W\in R^{512\times 117}$; for V-COCO [37], we use $W\in R^{512\times 24}$.

### 3.4. Training and Inference

#### 3.4.1. Training

Following UPT, we use focal loss [38] with a sigmoid layer to alleviate the imbalance between positive and negative samples and maintain numerical stability [20]. The specific formula is as follows:(9)$$\hat{y}\;=\;s_{h}\cdot s_{o}\cdot \sigma \left({\tilde{s}}_{k}\right)$$
(10)$$\mathrm{FL}\left(\hat{y},y\right)=\left\{\begin{array}{cc} -\beta {\left(1-\hat{y}\right)}^{\gamma }\log\left(\hat{y}\right), & y=1\\ -\left(1-\beta \right)\;{\hat{y}}^{\gamma }\log\left(1-\hat{y}\right), & y=0 \end{array}\right.$$
where $s_{h}$ and $s_{o}$ are the confidence scores of humans and objects, respectively. σ is the sigmoid function, $\sigma \left({\tilde{s}}_{k}\right)\in \left[0,1\right]$ is the normalized action score, and $\hat{y}\in \left[0,1\right]$ is the final action score. y∈{0,1} is the binary ground truth label. β∈[0,1] and $\gamma \in R_{+}$ are hyperparameters.

#### 3.4.2. Inference

Based on previous research [5], we excluded invalid combinations of actions and objects, such as holding an airplane. After SCG [21], we implemented an instance suppression strategy [20,31] (as shown in Figure 5) to avoid excessive object scores dominating the final HOI scores. Specifically, the object scores are reduced by adding a power of 2.8, and the final HOI scores $s_{k}$ are calculated as follows:(11)$$s_{k}\;=\;{\left(s_{h}\right)}^{\lambda }\cdot {\left(s_{o}\right)}^{\lambda }\cdot \sigma \left({\tilde{s}}_{k}\right)$$

## 4. Experiments

Section 4.1 details our experiment settings. Section 4.2 compares the results of our method with the state-of-the-art methods. Section 4.3 presents our model’s ablation study and comparative study. Section 4.4 demonstrates the costing study. Section 4.5 compares the expressive power of CNN features and transformer features. Section 4.6 presents qualitative results and limitations. If not specified, the models in this section use the ResNet-50 backbone and report their performance on the HICO-DET [5] dataset.

### 4.1. Experiment Settings

#### 4.1.1. Datasets

The HICO-DET dataset contains 37,633 valid training images and 9546 valid testing images, covering 80 objects, 117 actions, and 600 interactions. The V-COCO [37] dataset contains 4969 valid training images and 4532 valid testing images, covering 29 actions. Following UPT [20], we report results for 24 actions in the V-COCO dataset.

#### 4.1.2. Evaluation Metrics

We report the mean Average Precision (mAP) on the HICO-DET and V-COCO datasets. A prediction is considered a true positive if the intersection over union (IOU) of the human and object boxes with the ground truth boxes is higher than 0.5, and the HOI class is correct. For HICO-DET, we evaluate mAP on default and known object settings. Each setting contains 3 HOI sets: full (600 HOIs), rare (138 HOIs, less than 10 training instances), and non-rare (462 HOIs, 10 or more training instances). For V-COCO, we evaluate the mAP of Scenario 1 and Scenario 2. For the cases without role annotations, Scenario 1 requires the role prediction to be null, while Scenario 2 ignores the role prediction.

#### 4.1.3. Implementation Details

For HICO-DET, we use the DETR model [20] fine-tuned on HICO-DET. For V-COCO, we use the DETR model [20] trained from scratch on MS COCO [39]. Our data augmentation techniques include random cropping, image scaling, and color dithering [11,13]. We collect instances with scores greater than 0.2 and keep a maximum of 15 humans and 15 objects. The hyperparameters of focal loss are the same as UPT [20]. Our model is trained for 20 epochs using the AdamW [40] optimizer. The initial learning rate is $10^{-4}$, and the learning rate is reduced to $10^{-5}$ at the 10th epoch. The training uses a NVIDIA Tesla V100 device with a batch size of 16.

### 4.2. Comparisons with State-of-the-Art Methods

Table 1 and Table 2 compares the performance of our PCT with other methods. Although we do not conduct studies on aspects such as data augmentation and loss functions, such studies may complement ours. Our PCT with ResNet-50 achieves 33.63 mAP on the HICO-DET dataset, comparable to state-of-the-art methods. Notably, the PCT outperforms some models [17,20,41] with more powerful backbone networks. Our PCT achieves excellent performance using only visual features, indicating that our model is efficient. On the V-COCO dataset, our PCT with ResNet-101 significantly improves (about 2 mAP) over ResNet-50. In Scenario 2, the former achieves 67.1 mAP performance, comparable to state-of-the-art methods.

We observed that the performance of PCT using ResNet-50 and PCT using ResNet-101 is similar on the HICO-DET dataset. Typically, features extracted by ResNet-101 are superior to those of ResNet-50. To analyze this phenomenon, Table 3 demonstrates the impact of backbone networks and instance suppression strategies on performance.

Without performing instance suppression (λ=1), PCT with ResNet-101 outperforms PCT with ResNet-50 by 0.38 mAP. As λ gradually increases to 2.8, the performance gap between these two models decreases to 0.16 mAP. Additionally, the instance suppression strategy proposed by TIN [31] also reduces the performance gap between these two models. Experimental results show that the instance suppression strategies narrow the performance gap between these two models.

### 4.3. Ablation and Contrast Studies

#### 4.3.1. Ablation Study

We show the results of the ablation study in Table 4. The interaction head of the baseline model is a transformer encoder layer. Compared to the baseline, adding a transformer decoder layer improves about 2 mAP, adding the CNN features of human–object pairs improves about 3.3 mAP, and the complete PCT model improves about 4 mAP. In addition, removing the encoder layer from the complete model degrades about 0.8 mAP.

Following the UPT [20], we show the effect of model components on the interaction scores. Since we freeze the object detector, different interaction heads will use the same object detection result. We divide the negative samples according to the baseline model. An interaction score less than 0.05 is defined as an easy negative sample, otherwise a hard negative sample. In Table 5, we explore the impact of each component on positive and negative samples by controlling variables.

Table 5 shows that adding CNN features increases the average score of positive samples by 0.0167 and degrades the average score of hard negative samples by 0.0152, widening the gap between positive and negative sample scores. Adding the transformer decoder layer improves the average score of positive samples by 0.0127, but hard negative samples have almost no effect. Therefore, both of our proposed components are valid. Finally, our complete model improves the average score of positive samples by 0.0332 and reduces the average score of hard negative samples by 0.0273, further enlarging this gap. This indicates that the two components are complementary.

#### 4.3.2. Contrast Study

We show the results of different interaction heads in Table 6. Our interaction head first uses a transformer decoder layer and then a transformer encoder layer (D.E.+E.N., 33.63 mAP). The interaction head (E.N.+D.E., 33.41 mAP) with the opposite order of our submodules drops about 0.2 mAP in performance than ours. We speculate that the decoder layer needs to perform attention and generate HOI representations, which is challenging to balance. In addition, the design is inflexible due to the long global feature migration distance. The interaction head composed of double transformer encoder layers (E.N.+E.N., 33.15 mAP) is about 0.5 mAP lower than ours, which indicates that global features provide effective HOI clues. Although the interaction head consisting of double transformer decoder layers (D.E.+D.E., 33.63 mAP) covers ours, the performance does not improve further.

### 4.4. Computational Costing Study

Our network structure is straightforward, but the feature extraction method may increase the model’s complexity. To clarify the complexity of our model, Table 7 compares the computational cost of our method with the classical one-stage and two-stage methods. In addition, Table 8 shows the computational cost of our practical components. All experiments were performed on a NVIDIA GeForce GTX 1080 with an average input image size of [3, 887, 1055].

Table 7 shows that our model (13.02 fps) only lags behind the one-stage method QPIC (13.78 fps) in inference speed but outperforms other HOI detectors. Both UPT and our method are two-stage transformer-based methods. Compared to UPT (13.24 M), our model (8.60 M) has fewer trainable parameters, higher FPS, and is only slightly higher than UPT in terms of FLOPs. As shown in Figure 6, although UPT has lower FLOPs than ours, it has more submodules, which affects the inference speed.

Table 8 shows that our key components add a small number of parameters and a slight FPS reduction. The main reason for the increased FLOPs is caused by the decoder layer, which performs cross-attention between global features and human–object pair features.

### 4.5. Comparing CNN and Transformer Features

To fairly compare the performance of CNN and transformer features in HOI detection, we designed three variant models based on UPT. These models follow the structure shown in Figure 7, where the backbone CNN and the cooperative layer (Coop.) can be replaced. The modified encoder (M.E.) adds spatial information compared to the vanilla encoder (E.N.).

Table 9 shows that CNN features outperform transformer features in all variant models. When the vanilla transformer encoder layer (E.N.) is used in the cooperative layer, CNN features (31.16 mAP) outperform transformer features (30.39 mAP). We attribute this to the better local detail and stability of CNN features. Notably, when the cooperative layer uses the modified transformer encoder layer (M.E.), CNN features (32.17 mAP) further improve performance, while transformer features (30.22 mAP) perform unfavorably. We attribute this to the modified transformer encoder providing the spatial information, but the transformer features cannot effectively exploit this spatial information. When we replace the backbone network with the more powerful ResNet-101, improvements are achieved using both CNN and transformer features. Moreover, transformers show strong adaptability [20,26] and expressiveness.

### 4.6. Qualitative Results and Limitations

Figure 8 shows how the PCT model works qualitatively. We visualize attention at the transformer decoder layer in the interaction head. Figure 8c shows attention using only transformer features. We observe that the related human–object pair and its closer human–object pairs are focused, and we speculate that global features provide implicit spatial clues. As for the model paying attention to neighboring human–object pairs, the reason is the use of self-attention between human–object pairs. Figure 8d shows the attention of the complete PCT model. Using the fusion features of CNN and transformer can focus on relevant areas of human–object pairs more fully. We attribute this to the CNN effectively capturing local features and detailed information.

Qualitative analysis of HOI detection. Figure 9 demonstrates that our model can accurately identify HOI categories. However, our model does not perform well when the image is strongly occluded (as shown in Figure 9b) because the object score affects the final interaction score. Furthermore, we show several failure examples of the model. In Figure 10a, the false detection is caused by similar spatial relations with the straddling action. In Figure 10b, false detection occurs due to strong occlusion. In Figure 10c, the interaction score is too low due to fewer training samples. In Figure 10d, not detecting the correct object leads to wrong predictions. In Figure 10e, semantic ambiguity is caused by ambiguous interaction actions. These qualitative analyses provide valuable clues for model optimization.

## 5. Conclusions

In this paper, we try to combine the advantages of two-stage and one-stage methods. Our PCT extracts CNN features through an object detector to improve feature representation. Meanwhile, it uses the transformer’s global features to provide contextual information. Our method achieves competitive performance with a simple and efficient architecture. Then, we fairly compare the performance of CNN and transformer features in HOI detection. The results show that CNN features are not inferior to transformer features. Our work may advance the study of feature fusion and lightweight networks for HOI detection. In addition, we speculate that the features generated by the object detector are more suitable for object classification than interaction classification. In future work, we plan to adopt a universal visual pre-trained model to provide robust and high-quality features. Our method relies on human–object pair features to predict interactions, so it cannot predict scenarios that do not involve objects (such as running or falling). In addition, incorrect classification of objects by the object detector leads to wrong HOI predictions.

## Figures and Tables

**Figure 1 entropy-26-00205-f001:**
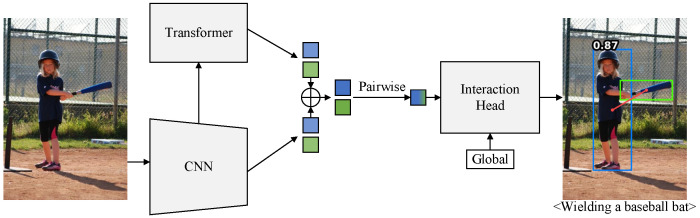
Our PCT pipeline.PCT is a two-stage transformer-based HOI detector. We use blue and green to represent humans and objects, respectively.

**Figure 2 entropy-26-00205-f002:**
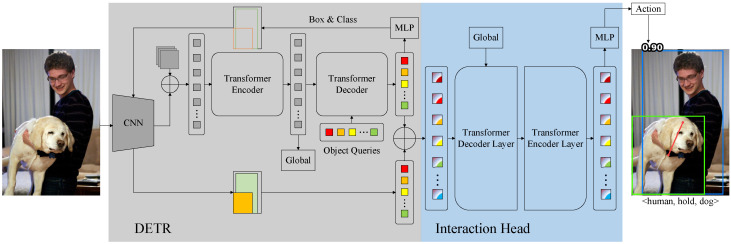
Our PCT framework comprises an object detector (DETR) and an interaction head. First, DETR detects humans and objects in the image, excluding instances with low confidence. Next, the retained humans and objects form human–object pairs through pairwise matching. Finally, the features of these human–object pairs and the global features are fed into the interaction head to predict action categories.

**Figure 3 entropy-26-00205-f003:**
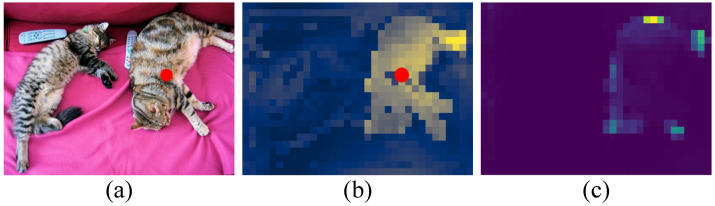
(**a**) Image with reference point. (**b**) Encoder self-attention for a reference point. (**c**) Decoder attention for an object query [35].

**Figure 4 entropy-26-00205-f004:**
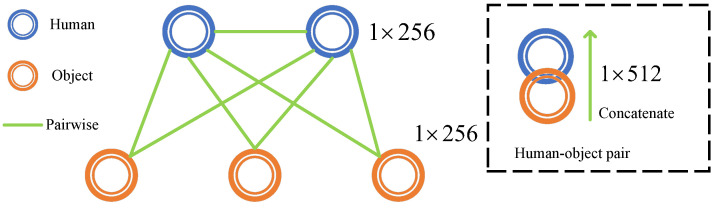
The process of pairwise matching between humans and objects.

**Figure 5 entropy-26-00205-f005:**
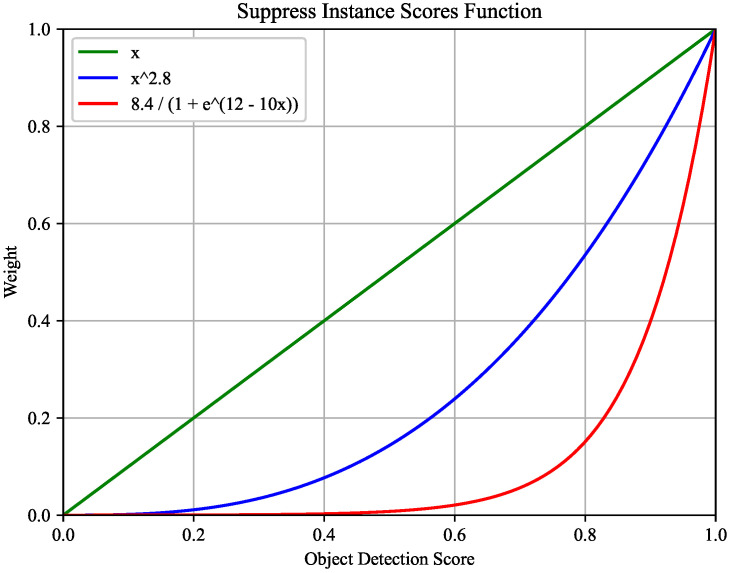
The green curve indicates no instance suppression fraction. The red and blue curves are the instance suppression strategies of TIN [31] and SCG [21], respectively. We follow the SCG.

**Figure 6 entropy-26-00205-f006:**
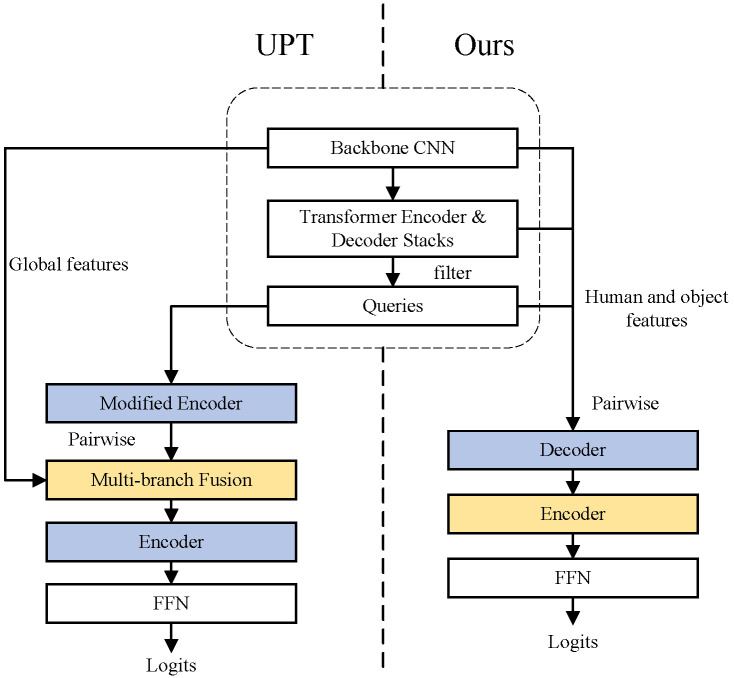
Comparison between UPT and our model.

**Figure 7 entropy-26-00205-f007:**
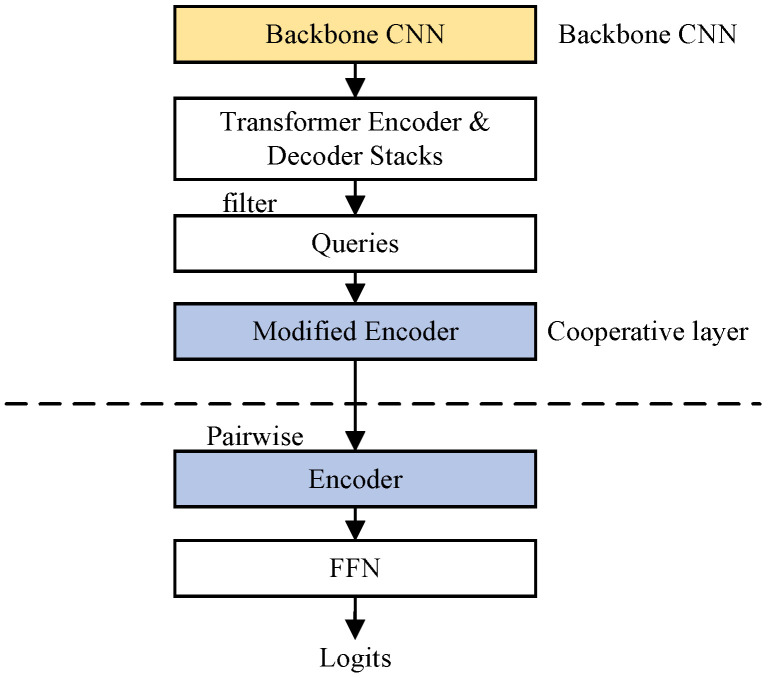
Framework of the UPT variant. The position before pairwise operations is called the cooperative layer.

**Figure 8 entropy-26-00205-f008:**
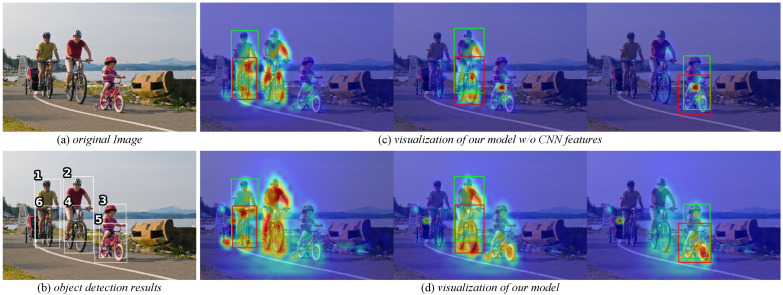
Effect of pairwise CNN features on attention. Please zoom in to view.

**Figure 9 entropy-26-00205-f009:**
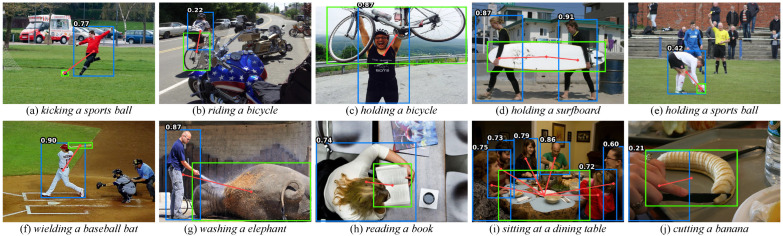
Successful cases. We filter out human–object pairs with scores lower than 0.2. Please zoom in to view.

**Figure 10 entropy-26-00205-f010:**
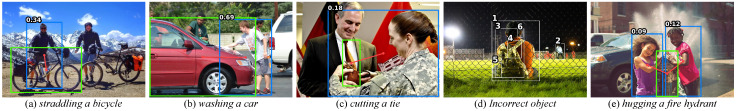
Failure cases. (**a**) Typical false detection. (**b**) Typical missed detection. (**c**) Low score. (**d**) Wrong object. (**e**) Semantic ambiguity.

**Table 1 entropy-26-00205-t001:** Performance (mAPx100) comparison on the HICO-DET dataset. The highest result for each setting is shown in bold.

Method	Backbone	Default Setting			Known Objects Setting		
		Full	Rare	Non-Rare	Full	Rare	Non-Rare
**CNN-based methods**							
HO-RCNN [5]	CaffeNet	7.81	5.37	8.54	10.41	8.94	10.85
InteractNet [7]	Res50-FPN	9.94	7.16	10.77	-	-	-
ICAN [6]	Res50	14.84	10.45	16.15	16.26	11.33	17.73
Xia et al. [42]	Res50	15.85	13.12	16.65	-	-	-
UnionDet [10]	Res50-FPN	17.58	11.72	19.33	19.76	14.68	21.27
VSGNet [18]	Res152	19.80	16.05	20.91	-	-	-
TIN [31]	Res50-FPN	20.93	18.95	21.32	23.02	20.96	23.42
PPDM [8]	Hourglass104	21.94	13.97	24.32	24.81	17.09	27.12
DRG [29]	Res50-FPN	24.53	19.47	26.04	27.98	23.11	29.43
SKGHOI [43]	Res50-FPN	26.95	21.28	28.56	-	-	-
SCG [21]	Res50-FPN	29.26	24.61	30.65	32.87	27.89	34.35
**Transformer-based methods**							
HOTR [12]	Res50	25.10	17.34	27.42	-	-	-
HOI-Trans [44]	Res101	26.61	19.15	28.84	29.13	20.98	31.57
AS-Net [23]	Res50	28.87	24.25	30.25	31.74	27.07	33.14
QPIC [11]	Res101	29.90	23.92	31.69	32.38	26.06	34.27
PhraseHOI [45]	Res101	30.03	23.48	31.99	33.74	27.35	35.64
MSTR [46]	Res50	31.17	25.31	32.92	34.02	28.83	35.57
CDN [17]	Res101	32.07	27.19	33.53	34.79	29.48	36.38
UPT [20]	Res101	32.31	28.55	33.44	35.65	31.60	36.86
Iwin [41]	Res101-FPN	32.79	27.84	**35.40**	35.84	28.74	36.09
PR-Net [47]	Res101	32.86	28.03	34.30	-	-	-
Ours (PCT)	Res50	33.63	28.73	35.10	36.96	31.47	**38.60**
Ours (PCT)	Res101	**33.79**	**29.70**	35.00	**37.16**	**32.63**	38.52

**Table 2 entropy-26-00205-t002:** Performance (mAPx100) comparison on the V-COCO dataset. The highest result for each setting is shown in bold.

Method	Backbone	${\mathrm{AP}}_{\mathrm{role}}^{S1}$	${\mathrm{AP}}_{\mathrm{role}}^{S2}$
**CNN-based methods**			
InteractNet [7]	Res50-FPN	40.0	-
ICAN [6]	Res50	45.3	-
Xia et al. [42]	Res50	47.2	-
UnionDet [10]	Res50-FPN	47.5	56.2
VSGNet [18]	Res152	51.8	57.0
TIN [31]	Res50-FPN	49.1	-
DRG [29]	Res50-FPN	51.0	-
SCG [21]	Res50-FPN	54.2	60.9
**Transformer-based methods**			
HOTR [12]	Res50	55.2	64.4
HOI-Trans [44]	Res101	52.9	-
AS-Net [23]	Res50	53.9	-
QPIC [11]	Res101	58.3	60.7
MSTR [46]	Res50	62.0	65.2
CDN [17]	Res101	**63.9**	65.9
UPT [20]	Res101	60.7	66.2
Iwin [41]	Res101-FPN	60.9	-
PR-Net [47]	Res101	62.9	64.2
Ours (PCT)	Res50	59.4	65.0
Ours (PCT)	Res101	61.4	**67.1**

**Table 3 entropy-26-00205-t003:** Results of implementing the instance suppression strategies.

	Instance Suppression Strategies			
**Backbone**	*λ* = 1.0	*λ* = 1.9	*λ* = 2.8	TIN [31]
Res50	32.10	33.44	33.63	32.92
Res101	32.48	33.63	33.79	33.11
Δ	+0.38	+0.19	+0.16	+0.19

**Table 4 entropy-26-00205-t004:** Ablation studies of our PCT components. The interaction head of the baseline model uses a standard encoder layer.

Model	CNN Features	Decoder	Full	Rare	Non-Rare
Baseline	-	-	29.48	24.46	30.97
	-	✔	31.50	26.98	32.84
	✔	-	32.80	27.89	34.26
Ours	✔	✔	**33.63**	**28.73**	**35.10**

**Table 5 entropy-26-00205-t005:** Change in average interaction score when a component is added to the reference network. According to the prediction results of the baseline model, the samples are categorized as positive, easy negatives, and hard negatives. The number of samples in each category is in parentheses.

		Δ Pos.	Δ Easy Neg.	Δ Hard Neg.
Reference	Δ Submodules	(25,342)	(4,373,390)	(46,172)
PCT w/o encoder layer	+ encoder layer	+0.0064	−0.0000	−0.0025
PCT w/o decoder layer	+ decoder layer	+0.0127	−0.0000	+0.0001
PCT w/o CNN features	+ CNN features	+0.0167	−0.0001	−0.0152
Baseline	PCT	**+0.0332**	−0.0001	**−0.0273**

**Table 6 entropy-26-00205-t006:** Comparison of results for different interaction heads. The acronym E.N. stands for encoder layer, and the acronym D.E. stands for decoder layer. The order of the acronyms indicates the order of the submodules.

Interaction Head	Full	Rare	Non-Rare
Only E.N.	32.80	27.89	34.26
Only D.E.	32.84	28.48	34.14
E.N.+E.N.	33.15	28.03	34.68
E.N.+D.E.	33.41	28.64	34.84
D.E.+D.E.	**33.63**	28.46	**35.18**
D.E.+E.N. (ours)	**33.63**	**28.73**	35.10

**Table 7 entropy-26-00205-t007:** Computational cost comparison. DETR is the object detector in our method. The number of trainable parameters is shown in parentheses.

Method	Backbone	Params (M)	FLOPs (G)	FPS
SCG	Res50-FPN	54.16	130.90	11.98
DETR	Res50	41.52	91.73	13.90
	Res101	60.46	163.70	9.61
QPIC	Res50	41.46	91.82	13.78
	Res101	60.40	163.79	9.62
UPT	Res50	54.76 (13.24)	91.91	12.23
	Res101	73.70 (13.24)	163.97	8.84
PCT(ours)	Res50	50.12 (8.60)	92.47	13.02
	Res101	69.06 (8.60)	164.49	9.27

**Table 8 entropy-26-00205-t008:** Costs of our model components. The Params column shows the number of trainable parameters. Changes in the number of parameters are shown in parentheses. All methods use resnet50 as the backbone network.

Method	Params (M)	FLOPs (G)	FPS
UPT	13.24	91.913	12.23
Only E.N. (Baseline)	3.21 (+0.00)	91.773	13.41
+CNN	4.26 (+1.05)	91.777	13.18
+D.E.	7.55 (+4.34)	92.468	13.21
+all (ours)	8.60 (+5.39)	92.472	13.02

**Table 9 entropy-26-00205-t009:** Comparison of the results of CNN features and transformer features. The acronym E.N. stands for standard encoder layer, and M.E. stands for modified encoder layer. In addition, the cooperative layer does not use the dropout layer.

Feature	Coop.	Full	Rare	Non-Rare
transformer	E.N.	30.39	24.95	32.02
CNN	E.N.	**31.16**	27.12	32.37
transformer	M.E.	30.22	24.71	31.86
CNN	M.E.	**32.17**	26.61	33.82
transformer	M.E.&Res101	30.47	26.38	31.69
CNN	M.E.&Res101	**32.23**	27.65	33.60

## Data Availability

All data available upon reasonable request.

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
