# Peer review of "Pairwise CNN-Transformer Features for Human–Object Interaction Detection"

_entropy, 2024, doi:10.3390/e26030205_

Round 1

Reviewer 1 Report

Comments and Suggestions for Authors

The authors proposed to fuse CNN and Transformer features to better use both instance and contextual information in the HOI task and achieved competitive performance compared to existing methods.

The following are my comments:

1. (Line 79-88) In the contribution part, the 1st and 2nd items are repeated and should be reorganized. The computational cost and inference time should also be emphasized here.

2. (Line 207-210) Details need to be provided on the strategy to make m human-object pairs each represented by a 512-dim feature if n1 people and n2 objects are detected in the detection stage.

3. (Line 265-273) The difference in DETR for different evaluations makes the conclusions less convincing, the authors should consider using a unified model for the detection part.

4. (Line 287-295) The results in Table 3 are too close to make the conjecture that instance suppression helps bring close the performance gap between Res50 and Res101 backbones. More results on different instance suppressions are necessary to support this point.

Comments on the Quality of English Language

The manuscript is hard to read and needs extensive revision for English expression.

Reviewer 2 Report

Comments and Suggestions for Authors

The paper at hand proposes a two-stage human-object interaction (HOI) detection method, composed of both a convolutional neural network (CNN) feature extractor and a cascaded transformer architecture that both extract deep features from the image frame regarding the human and the object of interest. Then, the extracted features from both models are fused to detect and estimate the interaction. The proposed work is novel and focuses on a well-established and competitive challenge.

In general, the paper is well-written and well-organized, yet several improvements are highly required.

Comments

1) Line 89: The authors claim that they will provide the code of the proposed system. Please provide a link of the repository (i.e. GitHub) within the paper, otherwise remove the sentence.

2) Based on a recent survey in the field [1] the advent of transformers have raised a third type of HOI detection methods, called end-to-end methods. Hence, recent methods are classified as two-stage, one-stage and end-to-end ones. The authors could read and discuss within the introduction of their manuscript the above three categories as well as provide examples for each category.

[1] Antoun, Maya, and Daniel Asmar. "Human object interaction detection: Design and survey." Image and Vision Computing 130 (2023): 104617.

3) The contributions of the work are clear. Yet, they could link better with the results and the enhancements achieved by the proposed method.

For instance, in point 1 (line 79) it is stated: “We propose a simple and efficient two-stage method PCT …”. In which term is it efficient and simple? In terms of execution time or complexity? Table 7 indicated higher complexity (in FLOPs) and number of parameters.

Additionally, to the reviewer’s understanding point 2 refers to the ablation study presented in section 4.3.1, that could be added in this description to further enhance the readability and highlight the contributions of the manuscript.

4) In section 2.2, features fusion constitutes a distinctive research challenge with a plethora of methods. Please, also refer to multiscale feature fusion [2], fusion with two-stage training [3], etc.

[2] Liu, Yu, et al. "Multiscale feature interactive network for multifocus image fusion." IEEE Transactions on Instrumentation and Measurement 70 (2021): 1-16.

[3] Kansizoglou, Ioannis, Loukas Bampis, and Antonios Gasteratos. "Deep feature space: A geometrical perspective." IEEE Transactions on Pattern Analysis and Machine Intelligence 44.10 (2021): 6823-6838.

5) Abbreviations, even well-known, should be explained in their first appearance throughout the manuscript, e.g., CNN (line 26), FPN (line 34), DETR (line 42), etc.

6) The conclusions section is too short. The authors could elaborate more regarding the contributions, possible applications as well as ideas for further improvements. Also, some limitations of the current systems should be discussed here.

Round 2

Reviewer 2 Report

Comments and Suggestions for Authors

The authors have revised the manuscript all of my concerns.

I am also satisfied with the author's replies to the other reviewer's comments.

Hence, I suggest the publication of the manuscript.